# Analysis of the Mechanism of Acid Mine Drainage Neutralization Using Fly Ash as an Alternative Material: A Case Study of the Extremely Acidic Lake Robule in Eastern Serbia

Nela Petronijević [1,*], Dragana Radovanović [2], Marija Štulović [2], Miroslav Sokić [1], Gvozden Jovanović [1,*], Željko Kamberović [3], Srđan Stanković [4], Srecko Stopic [5] and Antonije Onjia [3]

1 Institute for Technology of Nuclear and other Mineral Raw Materials, Center for Metallurgical Technologies, 11000 Belgrade, Serbia
2 Innovation Centre of the Faculty of Technology and Metallurgy Ltd., 11000 Belgrade, Serbia
3 Faculty of Technology and Metallurgy, University of Belgrade, 11000 Belgrade, Serbia
4 Bundesanstalt für Geowissenschaften und Rohstoffe, 30655 Hannover, Germany
5 IME Process Metallurgy and Metal Recycling, RWTH Aachen University, Intzestraße 3, 52056 Aachen, Germany
* Correspondence: nela_petronijevic@yahoo.com (N.P.); g.jovanovic@itnms.ac.rs (G.J.)

**Abstract:** Acid mine drainage (AMD) is a waste from mining sites, usually acidic, with high concentrations of sulfates and heavy metal ions. This study investigates the AMD neutralization process using fly ash (FA) as an alternative material. Samples of FA from coal-fired power plants in Serbia ("Nikola Tesla" (EF) and "Kostolac" (KOST)) were analyzed and used. The results were compared with the treatment efficiency of commercial neutralization agent (NaOH). The alkaline nature of FA was the basis for use in the treatment process of the extremely acid Lake Robule (pH 2.46), located in the mining areas of eastern Serbia. The optimal S/L ratio for the AMD neutralization process determined for EF was 25 wt.%, and for KOST it was 20 wt.%. The mechanism of the neutralization process was analyzed using the ANC test and PHREEQC program. The element concentrations and pH values in solutions indicated that FA samples could neutralize Lake Robule with more than 99% of Al, Fe, Cu, Zn, and more than 89% of Pb precipitated. Formation of insoluble (oxy)hydroxide forms ($Fe^{3+}$ and $Al^{3+}$ ions) creates favorable conditions for co-precipitation of other trace metals (Cu, Zn, Ni, Pb, and Cd) from AMD, which is further enhanced by cation adsorption on FA particles. FA proved to be a more effective neutralization agent than NaOH due to its adsorption effect, while among the FA samples, KOST was more effective due to the aging process through the carbonization reaction. Using FA as an alternative material is a promising and sustainable method for treating AMD, with economic and environmental benefits.

**Keywords:** acid mine drainage (AMD); fly ash; AMD treatment; zero waste; reuse waste; safe discharge of waste; synergy of waste reuse

## 1. Introduction

Acid mine drainage (AMD) has been identified as a significant global environmental issue closely associated with mining operations. AMD treatment is complicated due to high acidity and concentrations of hazardous elements (such as Cu, Zn, Ni, and Pb), as well as large volumes generated rapidly once the AMD process begins. It demands significant investments in materials and resources [1–3]. Waste from various industries is increasingly being used to neutralize AMD, which has economic and environmental benefits. In an ideal world, a solution to any industrial problem is only sustainable if it is energy efficient, does not pollute the environment, is economically viable, and produces little or no waste [4,5]. Wastes and by-products from various industries, which typically contain calcium oxides and carbonates, such as fly ash (FA) from coal combustion, can be used as neutralization

agents in AMD treatment. FA from coal-fired power plants is used for a variety of purposes, including adsorption of exhaust fumes, sulfur gases, and NOx, pollution removal (metals, inorganic or organic components of wastewater, phosphates, fluoride, boron, phenols, pesticides), zeolite synthesis, mine filling, construction, and the cement industry [6–8]. FA is a voluminous, hazardous alkaline waste that is difficult and expensive to deposit safely, pollutes the environment via spontaneous leaching of hazardous components, and can affect human health. However, its advantages include the availability of a free alkaline material that can be used to neutralize AMD by altering pH, resulting in the precipitation of valuable metal that can be recovered. Using a similar approach based on the neutralization potential of FA, Shirin et al. [9] successfully treated AMD samples with FA samples collected from four thermal power plants in India. In Serbia, coal is mined from open pits in the Kolubara Mining Basin (75%) and the Kostolac Mining Basin (25%).

Serbia's contribution to global FA production is relatively small, at around 6 million tons per year from the combustion of 32 million tons of coal [10,11]. The cement industry uses only 2.7% of FA [12]. The artificial Lake Robule was formed by the accumulation of AMD from over a century of mining activity in the Western Balkans region. The lake water has a low pH of 2.46 and high concentrations of the ions sulfate (7.36 g/L), $Al^{3+}$ (990 mg/L), and total Fe ions (287 mg/L). Lake Robule is an international environmental issue because its water flows into the Danube basin's rivers and the Black Sea [13]. Due to the importance of the issue, numerous researchers have looked into acidic water treatment and rehabilitation at Lake Robule [1–17]. This group of authors previously investigated the possibility of treating Lake Robule water with flotation tailings from the Majdanpek mine as a waste product and alternative neutralization agent [17].

The goal of this research is to investigate and optimize the treatment of AMD with FA in order to achieve sustainable development principles such as the circular economy and environmental conservation through improved waste management [18]. The study included the characterization of two types of FA produced in Serbian thermal power plants (EF and KOST) to investigate their use as neutralization agents and optimize AMD treatment in terms of S/L ratio and treatment time based on metal removal efficiency. Furthermore, the metal removal mechanism was determined by analyzing the resulting solutions and solid residues and thermodynamic modeling by the PHREEQC program. PHREEQC is a program that calculates the saturation index for all possible minerals of selected ions (in the database) that can form in solution as a function of time, pH conditions, and concentration. Finally, metal removal mechanism and efficiency were compared with the results of experiments using a conventional neutralization reagent, NaOH, whose effectiveness has been demonstrated [19,20] but which raises treatment costs [20–23], as well as the possibility of hazardous waste sludge formation [24].

## 2. Materials and Methods

### 2.1. Water Samples from the Lake Robule (AMD)

Lake Robule water was sampled for laboratory testing from a lake outlet pipe. Ten samples were collected directly from the stream using sterile 1-L containers made from high-density polyethylene. Prior to sampling, the containers were flushed with concentrated $HNO_3$ (Tehnohemija, Belgrade, Serbia) and deionized water. The procedure was repeated three times. The samples were refrigerated at 4 °C [17].

Analyses of selected chemical parameters were conducted by Inductively Coupled Plasma Optical Emission Spectroscopy (ICP-OES), using a Spectro Genesis spectrometer (Spectro Analytical Instruments, Kleve, Germany). Sulfate concentrations were determined gravimetrically with $BaCl_2$. Carbonates and hydrogen carbonates were tested voa titration, with 0.1 M HCl and phenolphthalein and methyl orange (Tehnohemija, Belgrade, Serbia) as indicators.

### 2.2. Fly ash Samples

Fly ash samples from the Nikola Tesla Thermal Power Plant (EF) in Obrenovac were sent to the ITNMS (Institute for Technology of Nuclear and Other Mineral Raw Materials) laboratory. Samples from the Drmno Thermal Power Plant in Kostolac (KOST) were collected in situ. The sampling method was "chessboard" [25], using a sterile scoop to collect samples from 25 points spaced 50 m apart at a depth of 1 m. The sample volume was 0.04 m$^3$. The samples were stored in sterile plastic bags.

A schematic geographic map of Serbia with the location of power plants Nikola Tesla and Drmno Kostolac, as well as Lake Robule, is given in Figure 1.

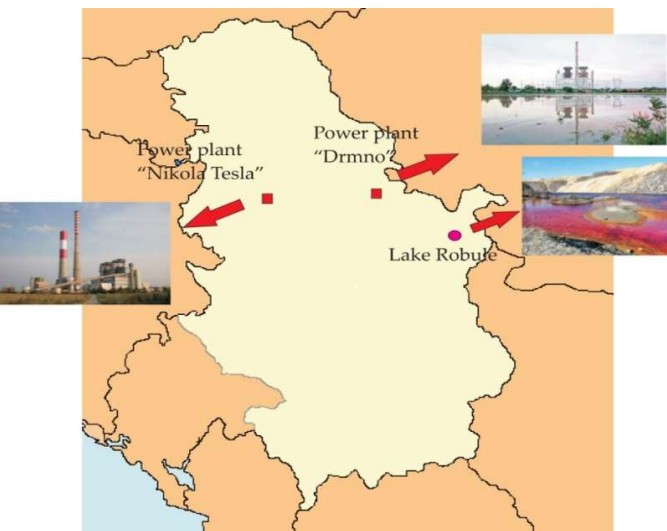

**Figure 1.** A schematic geographic map of Serbia showing the locations of the Nikola Tesla and Drmno Thermal power plants, as well as Lake Robule.

The chemical composition of the FA samples and solid residues after the treatment was determined by first dissolving the sample material in aqua regia and then measuring the concentrations of various elements by Atomic Absorption Spectroscopy (AAS) on an AAnalyst 300 (PerkinElmer, Norwalk, CT, USA).

The mineral composition of the fly ash and solid residues after the treatment was determined by and X-Ray Diffraction (XRD) (Zeiss-Jena, Oberkochen, Germany). A PW-1710 automated diffractometer (Philips) with a Cu tube operated at 30 mA and 40 kV (Federal Institute for Materials Research and Testing, Berlin, Germany) was used to obtain XRD patterns [17].

### 2.3. Defining the Optimal Solid/Liquid Ratio for AMD Treatment

A series of experiments were undertaken to determine the neutralizing properties of FA from power plants Nikola Tesla (EF) and Kostolac (KOST) in AMD treatment. The objective was to determine the optimal solid-to-liquid ratio required to neutralize AMD. Nine 100 mL Erlenmeyer flasks were used. Each contained the same amount of water, 50 mL, but with varying concentrations of the solid phase: 3%, 5%, 10%, 15%, 20%, 25%, 30%, and 40% by mass, resulting in different pulp densities. The flasks were placed in an orbital shaking incubator Heidolph Unimax 1010 (Heidolph, Schwabach, Germany), at 25 °C and 250 rpm. After two hours of agitation, the samples were left to rest at room temperature for seven days. The solid and liquid phases of the samples were separated by filtration, and the pH level was measured to determine the optimal EF and KOST required to achieve a neutral pH of treated water (the so-called representative sample).

### 2.4. AMD Treatment with FA Samples

The previously determined optimal S/L ratio for each FA sample was used in treatment experiments.

The effects of EF and KOST cation precipitation were analyzed by measuring the concentrations of the elements and pH over time. The solutions were prepared in three batches of nine Erlenmeyer flasks each. The flasks were then placed in the orbital shaking incubator with the same settings as in the previous experiment. After shaking, the solutions were left to rest for 30 min for the particles to be evenly distributed. The solutions were sampled after 5, 10, 15, and 30 min and after 2, 4, 24, 96, and 168 h. These experiments were made in triplicate. The samples were filtered using filter paper and analyzed by ICP-OES.

### 2.5. AMD Treatment with NaOH as a Conventional Neutralization Reagent

To define the reaction mechanism and compare the levels of AMD treatment efficiency (with FA EF and KOST), water from Lake Robule was neutralized with sodium hydroxide NaOH (Tehnohemija, Belgrade, Serbia), which is a conventional neutralization reagent. Three tests were set up, and each included the same volume of AMD sample, to which NaOH was gradually pipetted, and the pH level measured. Sodium hydroxide was added until the pH reached 7. All three assays were analyzed, and the results presented average metal concentrations.

### 2.6. Acid Neutralization Capacity Test

The Acid Neutralization Capacity test (ANC) is usually applied to treated waste to determine its ability to maintain stability in acidic environmental conditions [26]. In this paper, FA samples were subjected to the ANC test to determine the mechanism of acid neutralization and mineral phases contained in the FA, which would be involved in treating AMD. The ANC test was conducted according to the method described by Stegemann and Cote [27]: the FA samples (both initial and solid residues after treatment) were leached with nitric acid (Tehnohemija, Belgrade, Serbia) solution whose concentration gradually increased from 0 (distilled water) to 2 M in a series of 11 experiments (liquid to solid ratio (L/S) of 10). After 48 h of rotation, followed by centrifugation of the samples, the pH of the resulting leachates was measured and plotted against the equivalent amount of acid added per kg of FA (eq H$^+$/kg FA). The same procedure was applied to FA samples after the AMD treatment to determine the effects of the treatment on the neutralization capacity of the used FA samples.

### 2.7. Thermodynamic Modeling

PHREEQC is a well-known model primarily utilized in aqueous geochemical calculations [28]. PHREEQC is frequently used to make geochemical predictions between equilibrated solids, liquids, and gases. PHREEQC is capable of simulating kinetically controlled reactions, performing speciation and saturation-index calculations, transport calculations, and many other types of reaction calculations using the appropriate databases. By resolving equations based on the law of mass action (LMA) at a specific temperature and pressure, it performs calculations for thermodynamic equilibrium. It performs calculations for thermodynamic equilibrium to determine whether a phase is likely to dissolve or precipitate. Specifically, the analysis is based on identifying the solid phases that are predicted to form and precipitate by thermodynamics.

In this study, the saturation index was analyzed for minerals based on the chemistry of solutions after the neutralization treatment of AMD by FA.

Results of water analyses were input into PHREEQC and the program calculated the SI (Saturation Index) for all the possible minerals (in the database) that can be formed within the given components in the solution.

SI is defined as SI = log(IAP/Ksp), where IAP is the ion activity product, and Ksp is the equilibrium solubility product.

PHREEQC indicates that minerals are in stable form if the saturation index equals 0.0 (zero point zero). Following an initial analysis, if a solid phase has a negative SI, it is assumed to be under-saturated and will dissolve into the solution. Conversely, if the SI is positive, the solution is over-saturated, and a solid may form.

Phases with SI between −2.0 (negative two) and 2.0 (positive two) are most likely to be the controlling minerals for constituent element solubility, with the first precipitation of the minerals (in similar forms, e.g., oxides, hydroxides, carbonates) with the smallest positive SI [29].

## 3. Results and Discussion

### 3.1. Characterization of the Water Samples from Lake Robule

The physical and chemical characteristics of the AMD from Lake Robule and the standard deviation (SD) of measurement are presented in Table 1. The columns show 13 parameters (physical properties, ions, and elements), corresponding units, and measured values. Elements such as Se, Sb, Ba, Co, Cr, As, B, Ag, Be, and V were measured, but the values were below the detection limit.

**Table 1.** Physical and chemical properties of AMD from Lake Robule.

| Parameter | Results ± SD | Parameter | Results ± SD |
|---|---|---|---|
| Temperature °C | 7.0 ± 0.5 | pH | 2.46 ± 0.027 |
| Color and Odor | Yes/No | Eh | 615 ± 8.7 |
| $HCO_3^-$ (mg/L) | 0 | Zn (mg/L) | 17.5 ± 0.1 |
| $SO_4^{2-}$ (g/L) | 7.36 ± 0.703 | Pb (mg/L) | 0.19 ± 0.003 |
| $Fe^{3+}$ (mg/L) | 278.9 ± 11.2 | Mn (mg/L) | 65.9 ± 1.77 |
| $Fe^{2+}$ (mg/L) | 0.01 ± 0.004 | $Al^{3+}$ (mg/L) | 1040 ± 37 |
| Fe (total) (mg/L) | 279 ± 11.2 | Mg (mg/L) | 1184 ± 59 |
| Cu (mg/L) | 65.9 ± 1.08 | Ca (mg/L) | 396 ± 7.5 |
| Ni (mg/L) | 0.61 ± 0.021 | Cd (mg/L) | 0.01 ± 0.004 |

Based on the results presented in Table 1, it can be concluded that the AMD accumulated in the artificial Lake Robule has an extremely low pH value of 2.46, which determines it as acidic mine water. The red color of AMD results from the high Fe concentration (279 mg/L), which is 99.99% in the ferric form ($Fe^{3+}$). Furthermore, sulfate ($SO_4^{2-}$) and aluminum ($Al^{3+}$) ions are observed to predominate.

### 3.2. Characterization of the Fly Ashes Samples (EF and KOST)

#### 3.2.1. Chemical Composition

The chemical composition of the FA samples collected from power plants Nikola Tesla Thermal Power Plant (EF) and KOSTOLAC (KOST) are presented in Table 2, respectively, with values of SD, while the results of the X-ray diffraction analysis are presented in Figure 2.

**Table 2.** Mineral and chemical analysis of fly ash samples EF and KOST.

| | Major Components % ± SD | | | | | | | | | | |
|---|---|---|---|---|---|---|---|---|---|---|---|
| **Sample** | **pH** | **SiO₂** | **Al₂O₃** | **Fe₂O₃** | **CaO** | **MgO** | **K₂O** | **Na₂O** | **TiO₂** | **SO₃** | **MnO** |
| EF | 12 ± 0.15 | 53 ± 1.37 | 24 ± 0.76 | 8.13 ± 0.33 | 6.62 ± 0.308 | 1.53 ± 0.119 | 1.14 ± 0.067 | 0.39 ± 0.024 | 1.04 ± 0.015 | 0.91 ± 0.032 | 0.06 ± 0.003 |
| KOST | 9.3 ± 0.17 | 45.3 ± 1.03 | 22.4 ± 0.46 | 8.99 ± 0.53 | 7.31 ± 0.459 | 1.75 ± 0.146 | 0.85 ± 0.04 | 0.33 ± 0.026 | 0.99 ± 0.046 | 1.24 ± 0.006 | 0.08 ± 0.003 |
| | Elements mg/kg ± SD | | | | | | | | | |
| **Sample** | **Ba** | **F** | **Cl** | **Cu** | **V** | **Ni** | **Cr** | **As** | **Pb** | |
| EF | 79.8 ± 5.32 | 62.7 ± 2.54 | 32.1 ± 2.18 | 24.8 ± 0.15 | 23.9 ± 2.07 | 18.1 ± 0.44 | 15.7 ± 0.87 | 11.7 ± 0.99 | 9.46 ± 0.214 | |
| KOST | 85.5 ± 0.25 | 7.3 ± 0.46 | 24.7 ± 1.52 | 34.6 ± 0.54 | 32.6 ± 1.09 | 16.5 ± 0.64 | 12.9 ± 1.04 | 24.2 ± 2.4 | 12 ± 0.351 | |
| | Elements mg/kg ± SD | | | | | | | | | |
| **Sample** | **Zn** | **Mo** | **Se** | **Sb** | **Ag** | **Hg** | **Cd** | **LOI \*** | | |
| EF | 6.32 ± 0.056 | 1.71 ± 0.054 | <2.3 | <1.2 | <1.0 | <0.05 | <1.2 | 4.4 ± 0.15 | | |
| KOST | 10.45 ± 0.029 | 1.0 ± 0.085 | <2.3 | <1.2 | <1.0 | <0.05 | <1.2 | 11.2 ± 0.46 | | |

\* Loss on ignition at 950 °C.

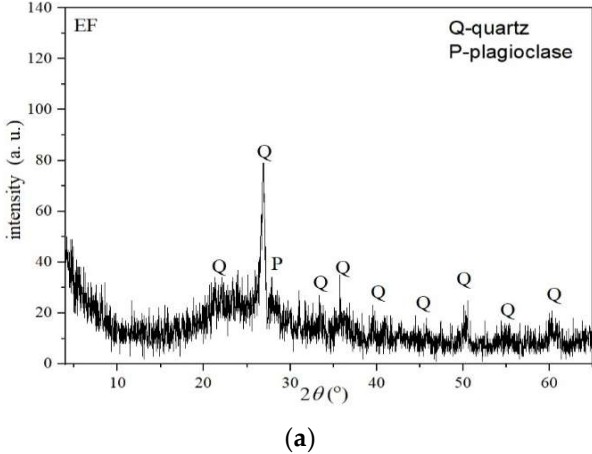
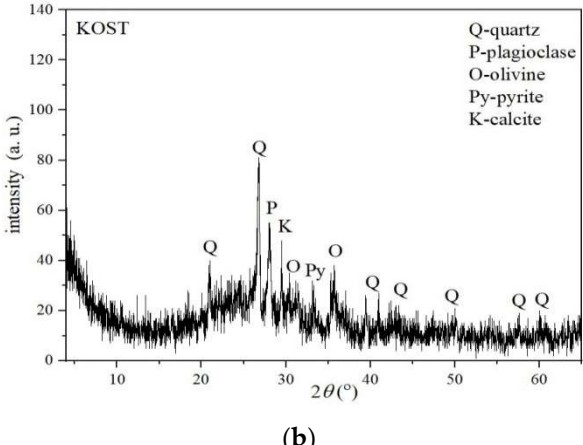

(**a**)                                                                                                           (**b**)

**Figure 2.** XRD diffractogram of fly ash samples: EF (**a**) and KOST (**b**).

Analyses of EF and KOST samples of FA revealed that $SiO_2$ and $Al_2O_3$ are the most prevalent oxides, while magnesium, calcium, aluminum, and iron are the most prevalent elements. According to the American standard ASTM C-618 (22), both ashes belong to the so-called F class of fly ash. This classification is based on the proportions of oxides $SiO_2$, $Al_2O_3$, and $Fe_2O_3$ and calcium oxide CaO, upon which the pozzolanic properties of fly ash depend. Class F consists of fly ash containing less than 10% calcium oxide and at least 70% other oxides.

### 3.2.2. XRD Analyses of Fly Ashes

Figure 2 presents the powder diffractograms of the FA samples EF and KOST. The *x*-ray diffraction determined the presence of quartz and plagioclase minerals in the polycrystalline powder samples of EF and KOST. At the same time, olivine, calcite, and traces of pyrite were only detected in the KOST sample. Chemical analysis also confirmed that $SiO_2$ is the most abundant oxide in the FA samples. In addition, XRD analysis revealed the predominance of amorphous matter, indicating a low degree of crystallinity.

### 3.2.3. ANC of Fly Ashes

ANC test results of FA samples (EF and KOST) with standard deviation values are presented in Figure 3. The EF sample has a very high initial pH value of 11.6, which indicates the presence of lime (CaO) [30], but its neutralization capacity depleted quickly to an [eq $H^+$/kg FA] value of 0.4. On the other hand, the KOST sample has a lower initial pH value of 9.0 and a gradual loss of neutralization capacity with a short plateau at pH 6. The plateau at these pH values is characteristic of calcite (CaCO3) [31], whose presence was confirmed by the mineralogical analysis of this sample. The variety in Ca-bearing compounds probably occurred due to the aging of the KOST sample at the landfill, which led to the carbonization through the reaction of the calcium oxide, initially present in the ash, with carbon dioxide [32,33].

The influence of various calcium compounds on ANC is evident in the experiments with 0.2 [eq $H^+$/kg FA], where the resulting solutions of EF and KOST samples have pH values of 7.5 and 6.5, respectively. In the next experiment, with an [eq $H^+$/kg FA] of 0.4, the pH value of the EF sample decreased significantly to 4.3, while the pH value of the KOST sample remained around 6, confirming the neutralization capacity of the present calcite. Finally, in the experiment with 0.6 [eq $H^+$/kg FA], when all the neutralization capacity of calcium compounds is consumed, both FA samples show almost identical pH values (3.9 for EF and 4.1 for KOST). After that, and until the end of the ANC test (2M $HNO_3$), both samples show a wide plateau around the pH value of 3. This result can be related to the dissolution of silicate minerals present in the FA samples. The dissolution process of silicate minerals, such as plagioclase, present in both FA samples, and olivine,

present in the KOST sample, consumes acid, allowing the material to possess neutralization capacity at these pH values [34–36].

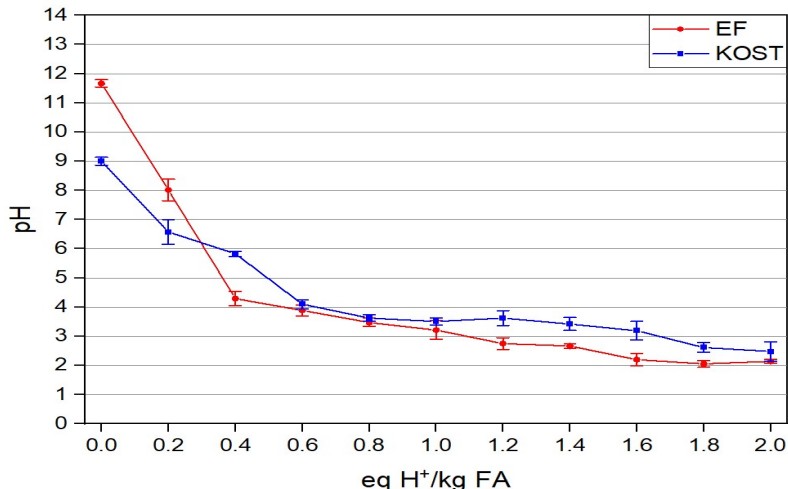

**Figure 3.** ANC test results of FA samples (EF and KOST).

Based on the results of the ANC test presented in Figure 3, it can be concluded that acid neutralization by FA involves two different processes. The first one is the reaction of acid neutralization by calcium compounds, such as lime and calcite, demonstrating a plateau at alkaline pH values during the ANC test, Reactions 1 and 2 [31,32,34,37]:

$$CaO + 2H^+ \leftrightarrow Ca^{2+} + H_2O \tag{1}$$

$$CaCO_3 + H^+ \leftrightarrow Ca^{2+} + HCO_3^- \tag{2}$$

The second process is the consumption of acid to dissolve aluminosilicate minerals, resulting in a plateau at acidic pH values during the ANC test [32,34], presented as Reaction 3:

$$CaAl_2Si_2O_8 + 8H^+ \leftrightarrow Ca^{2+} + 2Al^{3+} + 2H_4SiO_4 \tag{3}$$

In the works of Yanusa et al. [38] and Yan et al. [32], it was commented that the dissolution rate of aluminosilicate, compared to lime, is much slower, controlled by kinetics, and provides a long-term buffering effect to the material.

### 3.3. Results of AMD Treatment with Fly Ashes

#### 3.3.1. Optimal S/L Ratio

Results obtained during the determination of the optimal quantity of fly ash required for neutralization of the acid mine water samples to pH 7 are presented in Table S1 in the supplementary material.

The results presented in Table S1 in supplementary material show that when FA KOST is used to neutralize mine water, a pH value of 7.07 is obtained at a solid phase content of 10 g, i.e., at a pulp density of 20 percent, while the neutralization time is 1440 min (1 day). Following neutralization, the pH rose to 7.5 in 5760 min (4 days) and remained nearly constant at 7.44 after 10,080 min (7 days). Based on these findings, a pulp density of 20% was determined to be optimal for future experiments.

The optimal pulp density in tests where FA EF is used for neutralization is 25%. A pH of 7.16 was achieved in 10,080 min (7 days) with a solid phase concentration of 25%, or 12.5 g of sample in 50 mL of acidic wastewater, according to Table 3.

**Table 3.** Concentrations of metals as a function of time during the neutralization process of acid mine water with EF fly ash.

| Element (mg/L) | Time (min) | | | | | | | | | P * (%) ± SD |
|---|---|---|---|---|---|---|---|---|---|---|
| | 0 | 5 | 10 | 15 | 30 | 60 | 120 | 4320 | 10,080 | |
| pH | 2.46 ± 0.027 | 4.38 ± 0.077 | 4.62 ± 0.085 | 4.87 ± 0.105 | 5.4 ± 0.075 | 6.02 ± 0.07 | 6.19 ± 0.056 | 7 ± 0.024 | 7.1 ± 0.061 | - |
| Cu | 65.9 ± 1.08 | 19.6 ± 0.06 | 18 ± 0.37 | 15.5 ± 0.31 | 9.5 ± 0.24 | 0.3 ± 0.01 | 0.2 ± 0.01 | 0.1 ± 0.01 | 0.3 ± 0.01 | 99.2 ± 1.05 |
| Cd | 0.01 ± 0.001 | 0.05 ± 0.004 | 0.05 ± 0.004 | 0.05 ± 0.004 | 0.05 ± 0.004 | 0.05 ± 0.004 | 0.02 ± 0.003 | <0.007 | <0.005 | 50 ± 7.825 |
| Ni | 0.61 ± 0.021 | 0.69 ± 0.008 | 0.75 ± 0.012 | 0.78 ± 0.008 | 0.8 ± 0.025 | 0.79 ± 0.034 | 0.74 ± 0.023 | 0.66 ± 0.008 | 0.14 ± 0.007 | 77.05 ± 2.386 |
| Pb | 0.19 ± 0.003 | 0.12 ± 0.003 | 0.16 ± 0.001 | 0.36 ± 0.007 | 0.44 ± 0.015 | 0.21 ± 0.007 | 0.03 ± 0.001 | <0.02 | <0.02 | 89.47 ± 1.552 |
| Zn | 17.5 ± 0.1 | 18.9 ± 0.19 | 19.6 ± 0.13 | 20.4 ± 0.35 | 22.5 ± 0.19 | 18.6 ± 0.17 | 7.6 ± 0.01 | 3 ± 0.02 | 0.2 ± 0.02 | 99 ± 0.86 |
| Fe | 279 ± 11.2 | 2.3 ± 0.1 | 1.2 ± 0.08 | 1.2 ± 0.06 | 1.1 ± 0.02 | 1.1 ± 0.03 | <0.1 | <0.1 | <0.1 | 99 ± 0.7 |
| Al | 1040 ± 37 | 8.5 ± 0.3 | 8.1 ± 0.1 | 7.4 ± 0.1 | 7.3 ± 0.2 | 5.2 ± 0.1 | <0.1 | <0.1 | <0.1 | 99.9 ± 0.6 |
| Mg | 1184 ± 59 | 1196 ± 47 | 1207 ± 67 | 1202 ± 70.9 | 1253 ± 78.3 | 1200 ± 50 | 1249 ± 43.9 | 1174 ± 81.7 | 1190 ± 13.5 | +0.51 ± 0.05 |
| Ca | 396 ± 7.5 | 433 ± 14.7 | 428 ± 6.9 | 428 ± 9.2 | 435 ± 9.1 | 437 ± 14.4 | 476 ± 4.2 | 459 ± 13.1 | 439 ± 11.7 | +10.86 ± 0.3 |

* Precipitation.

### 3.3.2. Analysis of Solutions after the Treatment

For previously defined optimal samples of EF and KOST, kinetics tests were performed, and the concentrations of Al, Fe, Ca, Zn, Cd, Ni, Cu, Mg, and Pb in solution were measured during nine-time intervals (0 min, 5, 10, 15, 30, 60, 120, 4320, and 10,080 min). Tables 3 and 4 show the results of metal concentrations in the solution during the neutralization process with standard deviation values. All results presented in the tables are the average values of three measurements.

**Table 4.** Concentration of metals during the neutralization process of acid mine water with KOST fly ash.

| Element (mg/L) | Time (min) | | | | | | | | | P * (%) ± SD |
|---|---|---|---|---|---|---|---|---|---|---|
| | 0 | 5 | 10 | 15 | 30 | 60 | 120 | 4320 | 10,080 | |
| pH | 2.46 ± 0.027 | 5.11 ± 0.104 | 5.65 ± 0.055 | 6.02 ± 0.078 | 6.43 ± 0.051 | 6.63 ± 0.129 | 7.18 ± 0.078 | 7.33 ± 0.155 | 7.42 ± 0.125 | - |
| Cu | 65.9 ± 1.08 | 11.6 ± 0.24 | 8.8 ± 0.1 | 0.3 ± 0.01 | 0.1 ± 0.01 | 0.3 ± 0.01 | 0.2 ± 0.01 | 0.1 ± 0.01 | 0.2 ± 0.01 | 99.4 ± 0.47 |
| Cd | 0.01 ± 0.001 | 0.052 ± 0.002 | 0.046 ± 0.002 | 0.039 ± 0.0055 | 0.026 ± 0.001 | 0.013 ± 0.0012 | 0.008 ± 0.0006 | <0.007 | <0.005 | 50 ± 1.763 |
| Ni | 0.61 ± 0.021 | 0.58 ± 0.025 | 0.57 ± 0.016 | 0.58 ± 0.023 | 0.58 ± 0.03 | 0.53 ± 0.026 | 0.43 ± 0.021 | 0.32 ± 0.01 | 0.17 ± 0.006 | 72.13 ± 2.158 |
| Pb | 0.19 ± 0.003 | 0.04 ± 0.002 | 0.05 ± 0.003 | 0.03 ± 0.003 | 0.03 ± 0.003 | 0.03 ± 0.003 | 0.03 ± 0.003 | 0.03 ± 0.003 | <0.02 | 89.47 ± 1.481 |
| Zn | 17.5 ± 0.1 | 19.7 ± 0.12 | 14.3 ± 0.2 | 7.5 ± 0.13 | 5.6 ± 0.02 | 4.9 ± 0.04 | 1.6 ± 0.01 | <0.02 | <0.02 | 99.8 ± 0.54 |
| Fe | 279 ± 11.2 | 1.4 ± 0.1 | <0.1 | <0.1 | <0.1 | <0.1 | <0.1 | <0.1 | <0.1 | 99 ± 0.9 |
| Al | 1040 ± 37 | 1.3 ± 0.1 | 2.1 ± 0.1 | <0.1 | <0.1 | <0.1 | <0.1 | <0.1 | <0.1 | 99.9 ± 0.1 |
| Mg | 1184 ± 59 | 1189 ± 65.9 | 1163 ± 65.9 | 1237 ± 43.6 | 1206 ± 25.4 | 1197 ± 45.2 | 1154 ± 35.7 | 1140 ± 14.2 | 1139 ± 68.7 | 3.8 ± 0.2 |
| Ca | 396 ± 7.5 | 439 ± 21.6 | 433 ± 25.9 | 458 ± 12.4 | 443 ± 8.2 | 451 ± 20.9 | 471 ± 14.1 | 417 ± 1.6 | 434 ± 9.1 | +9.6 ± 0.4 |

* Precipitation.

The concentration of all elements, except for Ca and Mg, decreased during the neutralization procedure as the pH increased from 5.4 to 7.16. For certain elements, the highest precipitation was achieved at the end of the neutralization process, after 10,080 min (seven days) of the treatment. The highest value of precipitation (P) was 99% for Zn, Cu, Fe, and Al, followed by 89% for Pb, 76% for Ni, and 50% for Cd. During the treatment, the Ca and Mg concentrations in the solution increased, indicating the possibility of their leaching from the FA itself as a neutralizing agent.

The results of acid mine water neutralization with KOST FA revealed that the concentration of the observed metals decreased as the pH of the solution increased from 5.11 to 7.42. Cu, Zn, Fe, and Al exhibited the greatest decrease of 99.9%. It is observed that $Fe^{3+}$ ions precipitated abruptly at the first pH increase up to five, as well as the values of Al and relatively Cu, which had a sudden decrease in concentration up to pH 5 and were completely precipitated at pH 6. The concentration of Zn steadily grew until the pH of 5.11, and around pH 6.6, it started dropping until pH 7.33, when the Zn precipitation from the solution reached 99%. When it comes to Ni, its precipitation reached 72% at pH 7.42, and its plateau was more stable with a steeper precipitation decline when compared to EF FA treatment. Changes in Pb concentration showed the biggest difference between the two treatments. Both started with a steady decline until pH 4, but for the KOST treatment, it continued and showed a small rise around pH 5.65. Nevertheless, at pH 7.42, the precipitation of lead reached 89%, which is the same for the other treatments. Finally, the concentration of Cd rose during treatment until pH 6.63, when the precipitation started, and magnesium showed the lowest precipitation of 3.8%. As with EF FA neutralization,

during KOST FA treatment, the concentration of Ca in the solution increased, but in smaller amounts than with EF. This is probably due to the difference in composition of the FA, since KOST has more Ca in carbonized form than in the oxidized form, and its leachability is lower.

### 3.3.3. Analysis of Solid Residues after the Treatment

After the treatment, the precipitates have increased amounts of Al, Fe, Sb, Cu, Zn, Pb, Se, As, S, V, Cd, and Ni in FA samples (Table 5). This is in accordance with the concentration results from Tables 3 and 4, where 99% Al, Fe, Cu, and Zn have precipitated.

**Table 5.** Chemical composition of FA solid residues after AMD neutralization treatment.

| | Element Concentration (mg/kg) $\pm$ SD | | | | | | | | | |
|---|---|---|---|---|---|---|---|---|---|---|
| **Sample** | **Sb** | **Cu** | **V** | **Cd** | **Ni** | **Pb** | **Cr** | **Zn** | **S** | **Al** |
| EF7d | 28.2 $\pm$ 0.35 | 210 $\pm$ 0.4 | 142 $\pm$ 2.5 | 2.86 $\pm$ 0.215 | 102 $\pm$ 3 | 23.2 $\pm$ 0.31 | 113 $\pm$ 3.7 | 120 $\pm$ 1.2 | 3881 $\pm$ 287.6 | 121441 $\pm$ 4661.4 |
| KOST7d | 20.9 $\pm$ 0.41 | 304 $\pm$ 11 | 196 $\pm$ 3.6 | 1.35 $\pm$ 0.083 | 87 $\pm$ 1.1 | 67.8 $\pm$ 0.31 | 80 $\pm$ 2.8 | 171 $\pm$ 2.6 | 4833 $\pm$ 31.7 | 107691 $\pm$ 2574 |

| | Element Concentration (mg/kg) $\pm$ SD | | | | | | | | |
|---|---|---|---|---|---|---|---|---|---|
| **Sample** | **Fe** | **B** | **Co** | **Mn** | **As** | **Ba** | **Mo** | **Se** | **LOI \*** |
| EF7d | 45099 $\pm$ 2695.5 | <20 | 26.3 $\pm$ 0.82 | 391 $\pm$ 16 | 37.6 $\pm$ 0.33 | <100 | <0.5 | 25.4 $\pm$ 1.37 | 6.2 $\pm$ 0.1 |
| KOST7d | 58343 $\pm$ 1870.7 | <20 | 32.4 $\pm$ 0.89 | 605 $\pm$ 7.4 | 68.5 $\pm$ 2.32 | <100 | 1.34 $\pm$ 0.047 | 37.5 $\pm$ 1.64 | 15.8 $\pm$ 0.12 |

\* Loss on ignition at 950 °C.

### 3.3.4. XRD Analyses of the Solid Residues

The characteristic diffractograms of EF and KOST precipitates formed after 7 days of the neutralization procedure (EF7d and KOST7d) are shown in Figure 4.

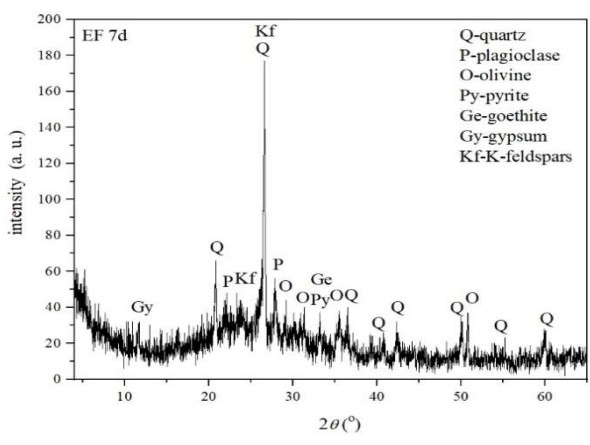

(**a**) XRD analysis of EF7d sample

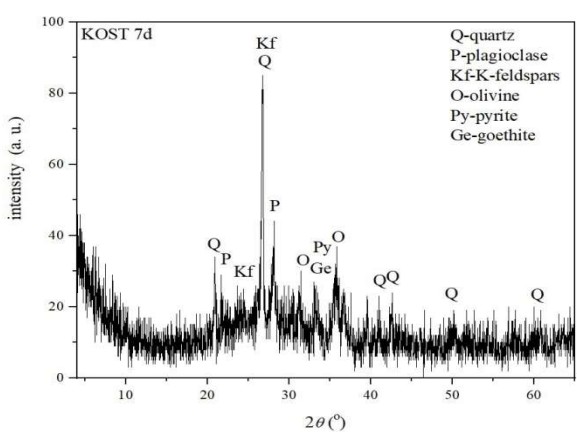

(**b**) XRD analysis of KOST7d sample

**Figure 4.** XRD analysis of EF7d and KOST7d fly ash samples.

Quartz remained the most abundant mineral in sample EF7d, followed by feldspars (plagioclases are more abundant than K-feldspars) and olivine. However, the presence of pyrite, goethite, and gypsum was also determined. The KOST7d sediment was found to contain quartz, plagioclase, K-feldspar, olivine, pyrite, and goethite, as determined by XRD. Quartz is the most abundant mineral, followed by feldspars (more plagioclase and less K-feldspar) and olivine, with pyrite and goethite being the least abundant. These results are nearly similar to those of an initial representative sample of EF and KOST analyzed by XRD prior to treatment, except for the occurrence of gypsum and goethite (Figure 2), formed by the acid neutralization process. The first one was formed by the transformation of calcite, and the second one was formed by the hydrolysis reaction of $Fe^{3+}$ ions from AMD.

### 3.3.5. ANC of the Solid Residues

Results of the ANC test of solid residues after the AMD treatment (samples EF 7d and KOST 7d) are presented in Figure 5. An apparent difference between FA samples before and after AMD treatment is the amount of acid that the material can neutralize until the pH value drops to a value of 7. These amounts were 0.24 and 0.16 [eq $H^+$/kg FA] for the EF and KOST samples, respectively (Figure 5). Both FA samples (EF 7d and KOST 7d) can neutralize 0.06 equivalent amount of acid per kg of material at a pH value of 7. This indicates significant consumption of mineral phases that would result in a pH value above 7 in the ANC test, such as lime and calcite, during the treatment. Initial pH values of 8.33 for the EF 7d sample and 7.87 for the KOST 7d sample may be the result of the presence of gypsum [26], formed by acid neutralization and confirmed by XRD analysis, Reactions (4) and (5).

$$CaO + 2H^+ + SO_4{}^{2-} \leftrightarrow CaSO_4 + H_2O \tag{4}$$

$$CaCO_3 + 2H^+ + SO_4{}^{2-} \leftrightarrow CaSO_4 + CO_2 + H_2O \tag{5}$$

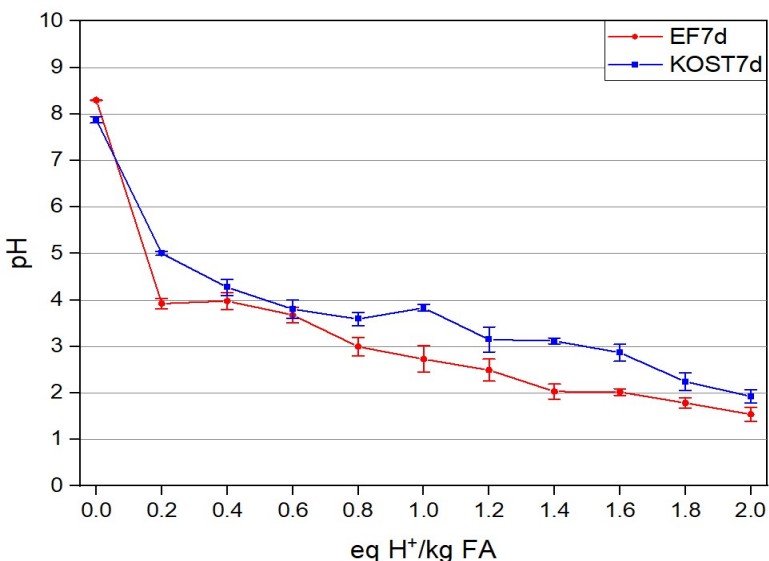

**Figure 5.** ANC test results of EF 7d and KOST 7d samples.

A plateau at pH values around 4, obtained for [eq $H^+$/kg FA] values between 0.2 and 0.6 (EF 7d) and between 0.4 and 1.0 (KOST 7d), can be attributed to the presence of goethite (FeOOH) formed during the treatment by the hydrolysis reaction of Fe3+ ions from AMD and with acid neutralization capacity at pH 3.5–4.0 [34,39], Reaction 6.

$$FeOOH + 3H^+ \leftrightarrow Fe^{3+} + 2H_2O \tag{6}$$

The expected plateau around pH 3 is shifted to lower pH values for both samples of solid residues. This indicates that aluminosilicate minerals were also consumed during AMD treatment through the dissolution reaction (Equation (3)).

### 3.4. Mechanism of the Metals Removal from the AMD

### 3.4.1. Results of PHREEQC Modeling

Program PHREEQC is used to calculate the saturation index for all the possible minerals (in the database) that can be formed based on given components in the solution as a function of time, pH condition, and concentration of elements in solutions for different neutralization process times (Tables 3 and 4). Results are shown in Tables 6 and 7.

**Table 6.** Results of the PHREEQC software simulation in the case with EF as a neutralization agent.

| Element | Mineral | t (Min)<br>pH | 0<br>2.46 | 5<br>4.38 | 10<br>4.62 | 15<br>4.87 | 30<br>5.4 | 60<br>6.02 | 120<br>6.19 | 4320<br>7 | 10,080<br>7.1 |
|---------|---------|---------------|-----------|-----------|------------|------------|-----------|------------|-------------|-----------|---------------|
| | | Formula | | | | | Saturation Index-SI | | | | |
| Al | $Al(OH)_3$ (a)<br>Gibbsite | $Al(OH)_3$<br>$Al(OH)_3$ | −5.80<br>−3.11 | −2.00<br>0.69 | −3.13<br>0.44 | −0.69<br>2.00 | 0.53<br>3.22 | 1.26<br>3.95 | −0.36<br>2.33 | −0.71<br>1.98 | −0.8<br>1.89 |
| Cd | $Cd(OH)_2$ | $Cd(OH)_2$ | −16.32 | −11.70 | −12.46 | −10.72 | −9.66 | −8.41 | −8.48 | −7.31 | −7.11 |
| Fe | $Fe(OH)_3$ (a)<br>Goethite | -<br>FeOOH | −9.37<br>−3.48 | −5.58<br>0.31 | −6.38<br>0.49 | −4.39<br>1.50 | −2.84<br>3.05 | −0.98<br>4.91 | −1.52<br>4.37 | 0.84<br>6.73 | 1.09<br>6.98 |
| Pb | $Pb(OH)_2$ | $Pb(OH)_2$ | −9.81 | −6.08 | −6.72 | −4.62 | −3.48 | −2.56 | −3.07 | 1.65 | 1.46 |
| Zn | $Zn(OH)_2$ (e) | $Zn(OH)_2$ | −10.74 | −6.73 | −7.48 | −5.72 | −4.62 | −3.46 | −3.51 | −2.29 | −3.27 |

**Table 7.** Results of the PHREEQC software simulation in the case with KOST fly ash neutralization agent.

| Element | Mineral | t (Min)<br>pH | 0<br>2.46 | 5<br>5.11 | 10<br>5.65 | 15<br>6.02 | 30<br>6.43 | 60<br>6.63 | 120<br>7.18 | 4320<br>7.33 | 10,080<br>7.42 |
|---------|---------|---------------|-----------|-----------|------------|------------|------------|------------|-------------|--------------|----------------|
| | | Formula | | | | | Saturation Index-SI | | | | |
| Al | $Al(OH)_3$ (a)<br>Gibbsite | $Al(OH)_3$<br>$Al(OH)_3$ | −5.80<br>−3.11 | −0.96<br>1.73 | 0.42<br>3.11 | −0.45<br>2.24 | −0.45<br>2.24 | −0.43<br>2.26 | −0.88<br>1.81 | −1.02<br>1.67 | −1.11<br>1.58 |
| Cd | $Cd(OH)_2$ | $Cd(OH)_2$ | −16.32 | −10.22 | −9.19 | −8.53 | −8.53 | −7.78 | −6.89 | −6.64 | −6.46 |
| Fe | $Fe(OH)_3$<br>Goethite | $Fe(OH)_3$<br>FeOOH | −9.37<br>−3.48 | −3.75<br>2.14 | −2.88<br>3.04 | −2.02<br>3.87 | −2.02<br>3.87 | −0.20<br>5.69 | 1.27<br>7.16 | 1.53<br>7.42 | 1.64<br>7.53 |
| Pb | $Pb(OH)_2$ | $Pb(OH)_2$ | −9.81 | −5.10 | −4.23 | −3.41 | −3.41 | −2.20 | −1.13 | 0.85 | 0.87 |
| Zn | $Zn(OH)_2$ (e) | $Zn(OH)_2$ | −10.74 | −5.25 | −4.31 | −3.86 | −3.86 | −2.82 | −2.20 | −3.80 | −3.63 |

For solution after treatment of AMD by EF, PHREEQC indicates that goethite (FeOOH) has a positive SI value after pH 4.38. Additionally, the mineral gibbsite ($Al(OH)_3$) has a positive SI value after 5 min at pH 4.38 and will precipitate. The software calculates the possibility of forming $Al(OH)_{3(a)}$ at pH 5.4, but the gibbsite formation is more likely to remove a majority of Al ions. Favorable conditions for Pb precipitation in $Pb(OH)_2$ form were after 4320 min at pH 7. There are no favorable conditions for Zn and Cd precipitation during this time. The precipitation of the other ions from the solution by the PHREEQC program was not recognized.

For solution after treatment of AMD by KOST, PHREEQC indicates that goethite (FeOOH) has a positive SI value from pH 5.11, which is very rapid after 5 min of treatment, and a solid may form. At the same time, the software calculated the possibility of $Fe(OH)_{3(a)}$ forming after pH 6.63. Mineral gibbsite $Al(OH)_3$ has positive SI values after 5 min at pH 5.11 and indicates precipitation of all Al ions from the solution. There are no favorable conditions for Cd precipitation in $Cd(OH)_2$ form and Zn in $Zn(OH)_2$ form during the treatment, but Pb can precipitate in $Pb(OH)_2$ form at pH 7.33. The precipitation of the other ions from the solution by the PHREEQC program was not recognized.

### 3.4.2. Treatment with NaOH

In order to compare alternative neutralization agents (EF and KOST samples of FA) with a conventional one, the experiment of AMD neutralization with NaOH was conducted. Three assays were set up, each with 50 mL of AMD sample, to which NaOH was gradually pipetted, and the pH level was measured. The pH was adjusted with sodium hydroxide until it reached 7. To achieve the pH of 7, the first assay required 810 µL of NaOH, the second 830 µL, and the third 800 µL of NaOH. The results of all three assays presented

average metal concentrations. The results of a neutralization treatment with the commercial neutralization agent NaOH and the standard deviation of measurement are shown in Table 8.

**Table 8.** Comparison of the metal concentration after neutralization treatment with fly ashes EF, KOST, and commercial NaOH.

| | Element Concentration (mg/L) | | | | | | |
|---|---|---|---|---|---|---|---|
| Sample | Cu | Fe | Al | Cd | Ni | Zn | Pb |
| AMD | $65.9 \pm 1.08$ | $279 \pm 11.2$ | $1040 \pm 37$ | $0.01 \pm 0.001$ | $0.61 \pm 0.021$ | $17.5 \pm 0.1$ | $0.19 \pm 0.003$ |
| EF | $0.3 \pm 0.01$ | <0.1 | <0.1 | <0.007 | $0.14 \pm 0.007$ | $0.2 \pm 0.02$ | <0.02 |
| KOST | <0.4 | <0.1 | <0.1 | <0.007 | $0.17 \pm 0.006$ | <0.02 | <0.02 |
| NaOH | $0.5 \pm 0.01$ | $0.18 \pm 0.007$ | $1.4 \pm 0.06$ | $0.1 \pm 0.013$ | $0.13 \pm 0.001$ | $0.4 \pm 0.01$ | $0.67 \pm 0.008$ |

Comparing the concentration of elements in acid water before and after neutralization treatment with both FA and a commercial agent, it is evident that KOST fly ash is the most effective. Both waste products in the form of fly ash have been shown to be superior to commercial NaOH in removing metals such as Al, Pb, and Cd, while for Ni, there was little difference. When removing Cu and Zn, treatment with EF fly ash was still better than NaOH, but both were far behind KOST fly ash neutralization. The noticeably higher concentrations of Fe, Al, Cd, Pb, and Cu were measured in solutions neutralized with NaOH, while they were below the detection limit in solutions neutralized with EF and KOST FA. The superiority of fly ash neutralizing agents is probably contributed by their ability to absorb metals on the surface of ash particles [40].

3.4.3. Co-Precipitation of Metals Cations

Iron and Aluminum

Fe and Al have the highest ion concentrations in AMD accumulated in Lake Robule: 279 mg/L of Fe (99% in $Fe^{3+}$ form) and 1040 mg/L of Al. Figure S1 in the Supplementary material shows the dependence of the Fe and Al metals concentrations on the pH value of the solutions during the AMD treatment. After the first 5 min of the treatment, when the pH of the solutions reached values of 4.38 for the EF sample and 5.11 for the KOST sample, the precipitations of these two elements already reached over 99%.

Precipitation of Fe and Al ions occurred due to the hydrolysis reaction at pH values from 3.0 to 3.5 for $Fe^{3+}$ and from 4.3 to 5.5 for $Al^{3+}$ [39] and the formation of insoluble iron and aluminum (oxy)hydroxides [41]. PHREEQC simulation results show that the thermo-dynamically most favorable precipitation of $Fe^{3+}$ ions is in the form of goethite ($\alpha$-FeOOH) and Al in the form of gibbsite ($\gamma$-Al(OH)$_3$). Compared to amorphous Fe and Al hydroxide, goethite and gibbsite cause a greater removal of $Fe^{3+}$ and $Al^{3+}$ from the solution due to their lower solubilities [34]. Goethite is formed by recrystallization of ferrihydrate through a dissolution–precipitation process under acidic conditions (pH < 4) and represents one of the most thermodynamically stable minerals within iron (oxy) hydroxide compounds [41]. Gibbsite is one of the crystalline forms of aluminum (oxy)hydroxide and is formed by the slow hydrolysis of Al under acidic conditions (pH < 6) [42]. Numerous studies have defined the role of Fe and Al (oxy)hydroxides in removing metals during the treatment of contaminated water of natural and industrial origin [41–43]. The retention of metal cations occurs via adsorption, mechanical occlusion, and structural incorporation during the formation of Fe and Al (oxy)hydroxides due to the nature of Fe and Al minerals and their specific bulk structures [41].

Copper and Zinc

The concentrations of Cu and Zn in AMD were 65.9 and 17.5 mg/L, respectively. A decrease in Cu and Zn concentration with increasing pH value during AMD treatment is shown in Figure S2 in the Supplementary material. The PHREEQC simulation did not show

thermodynamically favorable conditions for forming insoluble Cu compounds during the treatment. From the results shown, it can be concluded that the co-precipitation of Cu ions is associated with the formation and precipitation of Fe and Al (oxy)hydroxides. The treatment decreased Cu concentration to 0.3 mg/L at pH 6,02 and below the detection limit at pH 7. The mechanism of Cu retention can be described as the adsorption of $Cu^{2+}$ by surface sites of fly ash [37,44] and the adsorption and incorporation of $Cu^{2+}$ into the newly formed iron (oxy)hydroxide [41].

Zinc concentration was still high at pH values around 6, with 7.50 mg/L at pH 6.02 for the KOST sample and 7.6 mg/L at pH 6.19 for the EF sample. The concentration of Zn below the detection limit (0.01 mg/L) was reached at pH > 7.33 for the KOST sample. The reason for this is the relatively weak affinity of $Zn^{2+}$ for Fe and Al (oxy) hydroxide minerals [41]. The presented results indicated that retention of Zn could be related to adsorption and surface precipitation mechanisms onto silica or aluminosilicate sites of FA at pH 6–7 [37,44].

Nickel, Lead, and Cadmium

Ni, Pb, and Cd are metals with concentrations below 1.0 mg/L in AMD accumulated in Lake Robule: 0.61, 0.19, and 0.01 mg/L, respectively. Figure S3 in the Supplementary material presents the dependence of Ni, Pb, and Cd concentration on the pH value of solutions during the ANC treatment. With increasing pH values during the treatment, the concentration of Ni remains the same (for the KOST sample) or increases (for the EF sample). Dominant mechanism of Ni retention should be incorporated into goethite structure during Fe (oxy)hydroxide formation due to similar ionic radii of $Ni^{2+}$ and $Fe^{3+}$ [45]. However, a significant decrease in Ni concentration is achieved when the pH of the solution reaches values above 7, which may be the result of the formation of insoluble $Ni(OH)_2$.

The concentration of Pb decreases significantly at pH values around 6: 0.03 mg/L at pH 6.19 for the EF sample and at pH 6.02 for the KOST sample. This is consistent with the conclusions of Templeton et al. [46] that there is a significant increase in the affinity of $Pb^{2+}$ to be absorbed at the goethite surface at pH values above 6 [46]. The concentration of Pb is below the detection limit of 0.02 mg/L in solutions with pH values above 7. During the treatment, the concentration of Cd increases significantly until the pH value of solutions reaches 6, after which the Cd concentration gradually decreases. Similar to the Pb removal mechanism, the Cd concentration is below detection limits (0.005 mg/L) in solutions with pH above 7.

In Figures S2 and S3 in Supplementary material, it can be seen that, at the beginning of AMD treatment, there is an increase in the concentration of Zn, Ni, and Pb in solutions with EF FA and an increase in the concentration of Cd in solutions with both FA. This is due to their leaching from the mineral phases present in the FA samples upon contact with acidic AMD. Qureshi et al. [31] stated that the leaching of a certain element from FA depends on the solubility of the compound in which that element is present in the FA. Additionally, the majority (>70%) of the trace elements present in the FA are concentrated on the surfaces of the FA particles, making them more accessible to the leaching process. By comparing the results of applying two FA samples for AMD treatment, the leaching of Zn, Pb, and Ni from the EF sample and the absence of their leaching from the KOST sample can be observed. Similar to the results of the ANC test, this difference is an effect of the different mineralogical compositions of these two FA samples resulting from the aging process of the KOST sample. The aging of FA through the carbonization process leads to the transformation of soluble minerals, such as oxides, into less soluble species of these elements [30,32,33].

The mechanism of AMD treatment using FA can be defined by acid neutralization processes, resulting in an increase in the pH value of the solutions; the removal of $Fe^{3+}$ and $Al^{3+}$ ions as main metal ions in AMD chemical composition; and by the formation of insoluble (oxy)hydroxide compounds that lead to the co-precipitation of other trace metals

(Cu, Zn, Ni, Pb, and Cd) from AMD, with the additional effect of cation adsorption on FA particles.

## 4. Conclusions

Laboratory experiments confirmed that the fly ash could be applied in order to neutralize AMD accumulated in Lake Robule, and the following conclusions can be drawn:

- Due to their alkaline nature, both samples of fly ash EF and KOST can neutralize acidic mine waters at optimal solid liquid rations with increasing pH values.
- The optimal solid liquid ratio for EF fly ash is 25%, while for KOST fly ash it is 20%.
- In laboratory conditions, after neutralization with either EF or KOST fly ash, more than 99% of Al, Fe, Cu, and Zn and over 89% of Pb have precipitated.
- The removal of $Fe^{3+}$ and $Al^{3+}$ ions and formation of insoluble (oxy)hydroxide compounds, that occurs in first 5 min of neutralization, at pH 4.38 for the EF sample and 5.11 for the KOST sample, creates favorable conditions for co-precipitation of other trace metals (Cu, Zn, Ni, Pb, and Cd) from AMD, which is further enhanced by cation adsorption on FA particles.
- The neutralizing efficiency was determined by comparative analysis between the EF and KOST fly ash. The more effective neutralizing agent between these two FA was found to be KOST fly ash, since it can elevate the pH to the alkali range within a day due to changes in mineral phases. These changes in the mineral phase occurred due to the aging of FA through the carbonization process of calcium oxide, originally present in FA, with carbon dioxide, leading to different neutralization capabilities. ANC test results confirmed the presence of calcite minerals in KOST sample formed due to the aging of FA and the presence of gypsum in the ANC test of both solid residue samples formed through the neutralization process, as XRD results have already shown.
- Effects of the changes in pH values on the leachability of metal ions, and the neutralization mechanisms were confirmed by solution chemistry modeling results (PHREEQC software). The modeling results showed the predominant effect of the formation of Goethite and Gibbsite on the precipitation mechanics during the neutralization treatment with both FA.
- The application of NaOH for neutralization indicated that FA samples have the ability to absorb metals on the surface of ash particles, since they obtained better efficiency results than this commercial neutralization material.
- Applying EF or KOST fly ash to AMD leads to effective neutralization, making the AMD safe for disposal according to Serbian environmental laws and regulations.
- Applying EF and KOST FA as alternative neutralization materials can lead to economic and environmental benefits
- Analysis of the mechanism of acid mine drainage neutralization using fly ash as an alternative material in a case study of the extremely acidic lake Robule in Eastern Serbia represents great potential for transferable knowledge from the laboratory to the pilot plant in order to increase the valorization of metals present in aqueous solutions.

**Supplementary Materials:** The following supporting information can be downloaded at: https://www.mdpi.com/article/10.3390/w14203244/s1, Figure S1: Dependence of the Fe and Al concentration in solution on pH values during the AMD treatment; Figure S2: Dependence of Cu and Zn concentration in solution on pH values during the AMD treatment; Figure S3: Dependence of Ni, Pb and Cd concentration in solution on pH valuee during the AMD treatment; Table S1: Determination of the optimal quantity (S/L ration) of fly ash required for neutralization of the acid mine water samples to pH 7.

**Author Contributions:** Formulation of reasearch goals is done by N.P., Ž.K and S.S. (Srecko Stopic). while development of methodology by N.P., D.R. and S.S. (Srđan Stanković) Coordination of responsibilities for the research activity was planned by Ž.K. Conducting a research and investigation process was done by N.P and G.J. Software simulation and interpretation of data is performed by M.Š. and analyzis and synthesizis of data by D.R, while G.J. prepared manuscript gor publication.

Funding support is organized by M.S. and Ž.K. Original draft was wrtitten by N.P., D.R. and M.Š. and reviewed and edited by M.S., G.J. and S.S. (Srđan Stanković). Verification of the experiments setup and interpretation of results is done by S.S. (Srecko Stopic). and A.O. All authors have read and agreed to the published version of the manuscript.

**Funding:** The research presented in this paper was carried out with the financial support of the Ministry of Education, Science and Technological Development of the Republic of Serbia, within the funding of the scientific research work at the ITNMS and TMF, according to the contract with registration number 451-03-68/2022-14/200023 and 451-03-68/2022-14/200135.

**Institutional Review Board Statement:** Not applicable.

**Informed Consent Statement:** Not applicable.

**Data Availability Statement:** The data can be found within the manuscript.

**Conflicts of Interest:** The authors declare no conflict of interest.

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
