# Peer review of "Analysis of the Mechanism of Acid Mine Drainage Neutralization Using Fly Ash as an Alternative Material: A Case Study of the Extremely Acidic Lake Robule in Eastern Serbia"

_water, doi:10.3390/w14203244_

Round 1

Reviewer 1 Report

I enjoyed reading this comprehensive study. Please find my suggestions below.

1) Page 2, row 47, should it be "costs are low" and not high?

2) Page 3, row 124, is it "powder-XRD"?

3) Page 4, row 133, perhaps add "by mass" if correct (for %)?

4) Page 4, row 164, I would remove "(HNO3", also missing parenthesis.

5) Page 5, row 203, you write "such as", why not list all measured elements?

6) Page 5, Table 1. Why do you write > 304.5 for Mg and > 151.5 for Ca? How was it determined and why could you not dilute the sample and measure the concentration if it was too high?

7) Page 5, row 209. You write that sulfate and aluminium ions are observed to predominate. However, the concentration of several of the elements are higher than the conc. of sulfate. It is indeed the negatively charged ion of highest concentration. Perhaps rephrase the sentence?

8) Page 5, table 1, why not write the charge of the ions in the right column like is done in the left column?

9) Page 8, section 3.3.2. Did you do any repetitions, what is the repeatability of these experiments?

10) Page 10, row 315. Please remove "started with a" written two times.

11) Page 12, row 370. You write that time can be used in PHREEQC. However, did you use the program for kinetics? I think you only modelled thermodynamics? Please clarify.

12) Page 15, Figure 8. Can you add information about the time?

Reviewer 2 Report

The study used a waste product fly ash collected from two Serbian coal-fired power plants to treat acid mine drainage, confirming the effect of the neutralization material. The research was very simple, and the analyses are also simple.

1.     The “Abstract” section should be improved because we cannot get enough significant results.

2.     The third paragraph is composed of one sentence. Too short, is it okay?

3.     The figures of XRD results are a little blurry and KST should be KOST.

4.     The tables must be the three-line form.

5.     The physiochemical parameters of AMD had been published in a previous study. In this study, the authors described these data again, which was not suitable.

6.     Too many mirrors were in the study. For example, “and,” in Line 35 should delete “,”. “Concentration” in the title of Table 4 should be “Concentrations”.

7.     How to understand “Although they belong to the same FA class (class F) and have minor differences in chemical composition, the two investigated samples showed differences in the first stages of the ANC test.” in Lines 23-237?

8.     “where the resulting solution of an EF sample has pH 7.5 and KOST pH 6.5” in Lines 247-248 is grammatically wrong.

9.     The “Conclusions” section should be improved.

Reviewer 3 Report

This is an interesting manuscript on an important environmental matter. Please see the following comments

1) English language is not good. Please get advice from a fluent English speaker otherwise the paper cannot be published. see some examples

replace

 prevention and treat18 ment of its negative effects on the environment needs huge material investments.

with

appropriate treatment is needed in order to miminize its negative effects on the environment; this however necessitates important investments in materials and resources

replace

the AMD by altering the pH to certain levels

with

the AMD by lowering its pH to a certain effect

replace

Acid Mine Drainage (AMD), fly ash, amd treatment, zero waste, reuse waste, safe dis31 charge, synergy of waste reuse

with

Acid Mine Drainage (AMD), fly ash, AMD treatment, zero waste, , safe discharge of waste

etc...

2) the summary should be rewritten so that you dwell less on AMD (we know its problems) and more on the experimental design, the results and how they can be used

3) the same goes for the intrduction. many data and information should be removed to the discussion eg lines 55-63. so the introduction should be shortened and it should contain only description of the problem, that FA can be used because.... the aims and scope of your research and why it is important for an international audience

4) in Fig 1 please give title!

5) in materials and methods etc please give all chemical formulas correctly with subscripts

6) in materials and methods give in the same way (manufacturer, city, country of origin) all the instruments and reagents used

7) do not give any results in materials and methods and do not present results that you have not described the set up in the materials and methods section. This is very important-any results that have not been presented as methods should be erased

8) the analytical methods are very superficially written. Please see and quote 

Shirin et al Assessment of Characteristics of Acid Mine Drainage Treated

with Fly Ash, Applied Sciences

9) I understand that there are many results presented but you have 8 figures and 9 tables with results, this is extremely difficult for the reader to follow. please merge any tables etc that can be merged together. also eg for the reduction of metals in the AMD give only one Fig with the % reduction. some fig and tables should be in supplementary materials eg Table 9 fig 6 and 7

10) I am not quite sure why the experiment with NaOH was made and what you wanted to prove with this

11) use american English throughout eg favorable not favourable

12) the conclusions are very good but the discussion as stated before is very tiring and long for the reader and it mainly presents the results without much comparison to other studies. please a) shorten the discussion to 2/3 of present length b) erase many graphs and figures and /or transfer them to supplementray material c) give a clear qualitative explanation of the results d) enrich the comparison with relevant bibliography

Round 2

Reviewer 3 Report

the paper is acceptable. Please correct any english language mistakes with the editors

Author Response

Thank you for accepting our paper, we will correct any English mistakes with editors